# Block Coordinate Regularization by Denoising

**Yu Sun**
Washington University in St. Louis
sun.yu@wustl.edu

**Jiaming Liu**
Washington University in St. Louis
jiaming.liu@wustl.edu

**Ulugbek S. Kamilov**
Washington University in St. Louis
kamilov@wustl.edu

## Abstract

We consider the problem of estimating a vector from its noisy measurements using a prior specified only through a denoising function. Recent work on plug-and-play priors (PnP) and regularization-by-denoising (RED) has shown the state-of-the-art performance of estimators under such priors in a range of imaging tasks. In this work, we develop a new block coordinate RED algorithm that decomposes a large-scale estimation problem into a sequence of updates over a small subset of the unknown variables. We theoretically analyze the convergence of the algorithm and discuss its relationship to the traditional proximal optimization. Our analysis complements and extends recent theoretical results for RED-based estimation methods. We numerically validate our method using several denoiser priors, including those based on convolutional neural network (CNN) denoisers.

## 1   Introduction

Problems involving estimation of an unknown vector $x \in \mathbb{R}^n$ from a set of noisy measurements $y \in \mathbb{R}^m$ are important in many areas, including machine learning, image processing, and compressive sensing. Consider the scenario in Fig. 1, where a vector $x \sim p_x$ passes through the measurement channel $p_{y|x}$ to produce the measurement vector $y$. When the estimation problem is ill-posed, it becomes essential to include the prior $p_x$ in the estimation process. However, in high-dimensional settings, it is difficult to directly obtain the true prior $p_x$ for certain signals (such as natural images) and one is hence restricted to various indirect sources of prior information on $x$. This paper considers the cases where the prior information on $x$ is specified only via a denoising function, $\mathsf{D} : \mathbb{R}^n \to \mathbb{R}^n$, designed for the removal of additive white Gaussian noise (AWGN).

There has been considerable recent interest in leveraging denoisers as priors for the recovery of $x$. One popular strategy, known as plug-and-play priors (PnP) [1], extends traditional proximal optimization [2] by replacing the proximal operator with a general off-the-shelf denoiser. It has been shown that the combination of proximal algorithms with advanced denoisers, such as BM3D [3] or DnCNN [4], leads to the state-of-the-art performance for various imaging problems [5–15]. A similar strategy has also been adopted in the context of a related class of algorithms known as approximate message passing (AMP) [16–19]. Regularization-by-denoising (RED) [20], and the closely related deep mean-shift priors [21], represent an alternative, in which the denoiser is used to specify an explicit regularizer that has a simple gradient. More recent work has clarified the existence of explicit RED regularizers [22], demonstrated its excellent performance on phase retrieval [23], and further boosted its performance in combination with a deep image prior [24]. In short, the use of advanced denoisers has proven to be essential for achieving the state-of-the-art results in many contexts. However, solving the corresponding estimation problem is still a significant computational challenge, especially in the context of high-dimensional vectors $x$, typical in modern applications.

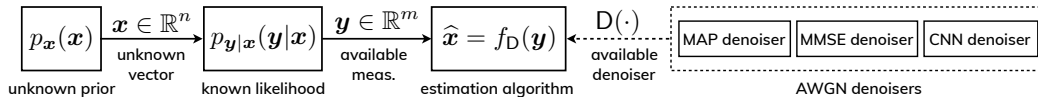

Figure 1: *The estimation problem considered in this work. The vector $\boldsymbol{x} \in \mathbb{R}^n$, with a prior $p_{\boldsymbol{x}}(\boldsymbol{x})$, passes through the measurement channel $p_{\boldsymbol{y}|\boldsymbol{x}}(\boldsymbol{y}|\boldsymbol{x})$ to result in the measurements $\boldsymbol{y} \in \mathbb{R}^m$. The estimation algorithm $f_{\mathsf{D}}(\boldsymbol{y})$ does not have a direct access to the prior, but can rely on a denoising function $\mathsf{D} : \mathbb{R}^n \rightarrow \mathbb{R}^n$, specifically designed for the removal of additive white Gaussian noise (AWGN). We propose block coordinate RED as a scalable algorithm for obtaining $\boldsymbol{x}$ given $\boldsymbol{y}$ and $\mathsf{D}$.*

In this work, we extend the current family of RED algorithms by introducing a new *block coordinate RED (BC-RED)* algorithm. The algorithm relies on random partial updates on $\boldsymbol{x}$, which makes it scalable to vectors that would otherwise be prohibitively large for direct processing. Additionally, as we shall see, the overall computational complexity of BC-RED can sometimes be lower than corresponding methods operating on the full vector. This behavior is consistent with the traditional coordinate descent methods that can outperform their full gradient counterparts by being able to better reuse local updates and take larger steps [25–29]. We present two theoretical results related to BC-RED. We first theoretically characterize the convergence of the algorithm under a set of transparent assumptions on the data-fidelity and the denoiser. Our analysis complements the recent theoretical analysis of full-gradient RED algorithms in [22] by considering block-coordinate updates and establishing the explicit worst-case convergence rate. Our second result establishes backward compatibility of BC-RED with the traditional proximal optimization. We show that when the denoiser corresponds to a proximal operator, BC-RED can be interpreted as an approximate MAP estimator, whose approximation error can be made arbitrarily small. To the best of our knowledge, this explicit link with proximal optimization is missing in the current literature on RED. BC-RED thus provides a flexible, scalable, and theoretically sound algorithm applicable to a wide variety of large-scale estimation problems. We demonstrate BC-RED on image recovery from linear measurements using several denoising priors, including those based on convolutional neural network (CNN) denoisers.

All proofs and some technical details that have been omitted for space appear in the full paper [30] that also provides more background and simulations.

## 2   Background

It is common to formulate the estimation in Figure 1 as an optimization problem

$$\widehat{\boldsymbol{x}} = \underset{\boldsymbol{x} \in \mathbb{R}^n}{\arg\min} f(\boldsymbol{x}) \quad \text{with} \quad f(\boldsymbol{x}) = g(\boldsymbol{x}) + h(\boldsymbol{x}), \tag{1}$$

where $g$ is the data-fidelity term and $h$ is the regularizer. For example, the maximum a posteriori probability (MAP) estimator is obtained by setting

$$g(\boldsymbol{x}) = -\log(p_{\boldsymbol{y}|\boldsymbol{x}}(\boldsymbol{y}|\boldsymbol{x})) \quad \text{and} \quad h(\boldsymbol{x}) = -\log(p_{\boldsymbol{x}}(\boldsymbol{x})),$$

where $p_{\boldsymbol{y}|\boldsymbol{x}}$ is the likelihood that depends on $\boldsymbol{y}$ and $p_{\boldsymbol{x}}$ is the prior. One of the most popular data-fidelity terms is least-squares $g(\boldsymbol{x}) = \frac{1}{2}\|\boldsymbol{y} - \boldsymbol{A}\boldsymbol{x}\|_2^2$, which assumes a linear measurement model under AWGN. Similarly, one of the most popular regularizers is based on a sparsity-promoting penalty $h(\boldsymbol{x}) = \tau\|\boldsymbol{D}\boldsymbol{x}\|_1$, where $\boldsymbol{D}$ is a linear transform and $\tau > 0$ is the regularization parameter [31–34].

Many widely used regularizers, including the ones based on the $\ell_1$-norm, are nondifferentiable. Proximal algorithms [2], such as the proximal-gradient method (PGM) [35–38] and alternating direction method of multipliers (ADMM) [39–42], are a class of optimization methods that can circumvent the need to differentiate nonsmooth regularizers by using the proximal operator

$$\mathsf{prox}_{\mu h}(\boldsymbol{z}) := \underset{\boldsymbol{x} \in \mathbb{R}^n}{\arg\min} \left\{ \frac{1}{2}\|\boldsymbol{x} - \boldsymbol{z}\|_2^2 + \mu h(\boldsymbol{x}) \right\}, \quad \mu > 0, \quad \boldsymbol{z} \in \mathbb{R}^n. \tag{2}$$

The observation that the proximal operator can be interpreted as the MAP denoiser for AWGN has prompted the development of PnP [1], where the proximal operator $\mathsf{prox}_{\mu h}(\cdot)$, within ADMM or PGM, is replaced with a more general denoising function $\mathsf{D}(\cdot)$.

Consider the following alternative to PnP that also relies on a denoising function [20, 21]

$$\boldsymbol{x}^t \leftarrow \boldsymbol{x}^{t-1} - \gamma \left( \nabla g(\boldsymbol{x}^{t-1}) + \mathsf{H}(\boldsymbol{x}^{t-1}) \right) \quad \text{where} \quad \mathsf{H}(\boldsymbol{x}) := \tau(\boldsymbol{x} - \mathsf{D}(\boldsymbol{x})), \quad \tau > 0. \quad (3)$$

Under some conditions on the denoiser, it is possible to relate $\mathsf{H}(\cdot)$ in (3) to some explicit regularization function $h$. For example, when the denoiser is locally homogeneous and has a symmetric Jacobian [20, 22], the operator $\mathsf{H}(\cdot)$ corresponds to the gradient of the following function

$$h(\boldsymbol{x}) = \frac{\tau}{2} \boldsymbol{x}^\mathsf{T} (\boldsymbol{x} - \mathsf{D}(\boldsymbol{x})). \quad (4)$$

On the other hand, when the denoiser corresponds to the minimum mean squared error (MMSE) estimator $\mathsf{D}(\boldsymbol{z}) = \mathbb{E}[\boldsymbol{x}|\boldsymbol{z}]$ for the AWGN denoising problem [21, 22], $\boldsymbol{z} = \boldsymbol{x} + \boldsymbol{e}$, with $\boldsymbol{x} \sim p_{\boldsymbol{x}}(\boldsymbol{x})$ and $\boldsymbol{e} \sim \mathcal{N}(\boldsymbol{0}, \sigma^2 \boldsymbol{I})$, the operator $\mathsf{H}(\cdot)$ corresponds to the gradient of

$$h(\boldsymbol{x}) = -\tau\sigma^2 \mathsf{log}(p_{\boldsymbol{z}}(\boldsymbol{x})), \quad p_{\boldsymbol{z}}(\boldsymbol{x}) = (p_{\boldsymbol{x}} * p_e)(\boldsymbol{x}) = \int_{\mathbb{R}^n} p_{\boldsymbol{x}}(\boldsymbol{z}) \phi_\sigma(\boldsymbol{x} - \boldsymbol{z}) \, \mathsf{d}\boldsymbol{z}, \quad (5)$$

where $\phi_\sigma$ is the Gaussian probability density function of variance $\sigma^2$ and $*$ denotes convolution. In this paper, we will use the term RED to denote *all* methods seeking the fixed points of (3). The key benefits of the RED methods [20–24] are their explicit separation of the forward model from the prior, their ability to accommodate powerful denoisers (such as the ones based on CNNs) without differentiating them, and their state-of-the-art performance on a number of imaging tasks. The next section further extends the scalability of RED by designing a new block coordinate RED algorithm.

## 3   Block Coordinate RED

All the current RED algorithms operate on vectors in $\mathbb{R}^n$. We propose BC-RED, shown in Algorithm 1, to allow for partial randomized updates on $\boldsymbol{x}$. Consider the decomposition of $\mathbb{R}^n$ into $b \geq 1$ subspaces

$$\mathbb{R}^n = \mathbb{R}^{n_1} \times \mathbb{R}^{n_2} \times \cdots \times \mathbb{R}^{n_b} \quad \text{with} \quad n = n_1 + n_2 + \cdots + n_b.$$

For each $i \in \{1, \ldots, b\}$, we define the matrix $\mathsf{U}_i : \mathbb{R}^{n_i} \to \mathbb{R}^n$ that injects a vector in $\mathbb{R}^{n_i}$ into $\mathbb{R}^n$ and its transpose $\mathsf{U}_i^\mathsf{T}$ that extracts the $i$th block from a vector in $\mathbb{R}^n$. Then, for any $\boldsymbol{x} = (\boldsymbol{x}_1, \ldots, \boldsymbol{x}_b) \in \mathbb{R}^n$

$$\boldsymbol{x} = \sum_{i=1}^b \mathsf{U}_i \boldsymbol{x}_i \quad \text{with} \quad \boldsymbol{x}_i = \mathsf{U}_i^\mathsf{T} \boldsymbol{x} \in \mathbb{R}^{n_i}, \quad i = 1, \ldots, b \quad \Leftrightarrow \quad \sum_{i=1}^b \mathsf{U}_i \mathsf{U}_i^\mathsf{T} = \mathsf{I}. \quad (6)$$

Note that (6) directly implies the norm preservation $\|\boldsymbol{x}\|_2^2 = \|\boldsymbol{x}_1\|_2^2 + \cdots + \|\boldsymbol{x}_b\|_2^2$ for any $\boldsymbol{x} \in \mathbb{R}^n$. We are interested in a block-coordinate algorithm that uses only a subset of operator outputs corresponding to coordinates in some block $i \in \{1, \ldots, b\}$. Hence, for an operator $\mathsf{G} : \mathbb{R}^n \to \mathbb{R}^n$, we define the block-coordinate operator $\mathsf{G}_i : \mathbb{R}^n \to \mathbb{R}^{n_i}$ as

$$\mathsf{G}_i(\boldsymbol{x}) := [\mathsf{G}(\boldsymbol{x})]_i = \mathsf{U}_i^\mathsf{T} \mathsf{G}(\boldsymbol{x}) \in \mathbb{R}^{n_i}, \quad \boldsymbol{x} \in \mathbb{R}^n. \quad (7)$$

We introduce the following BC-RED algorithm.

---
**Algorithm 1** Block Coordinate Regularization by Denoising (BC-RED)

---
1: **input:** initial value $\boldsymbol{x}^0 \in \mathbb{R}^n$, parameter $\tau > 0$, and step-size $\gamma > 0$.
2: **for** $k = 1, 2, 3, \ldots$ **do**
3:     Choose an index $i_k \in \{1, \ldots, b\}$
4:     $\boldsymbol{x}^k \leftarrow \boldsymbol{x}^{k-1} - \gamma \mathsf{U}_{i_k} \mathsf{G}_{i_k}(\boldsymbol{x}^{k-1})$
        where $\mathsf{G}_i(\boldsymbol{x}) := \mathsf{U}_i^\mathsf{T} \mathsf{G}(\boldsymbol{x})$ with $\mathsf{G}(\boldsymbol{x}) := \nabla g(\boldsymbol{x}) + \tau(\boldsymbol{x} - \mathsf{D}(\boldsymbol{x}))$.
5: **end for**

---

Note that when $b = 1$, we have $n = n_1$ and $\mathsf{U}_1 = \mathsf{U}_1^\mathsf{T} = \mathsf{I}$. Hence, the theoretical analysis in this paper is also applicable to the full-gradient RED algorithm in (3).

As with traditional coordinate descent methods (see [28] for a review), BC-RED can be implemented using different block selection strategies. The strategy adopted for our theoretical analysis selects block indices $i_k$ as i.i.d. random variables distributed uniformly over $\{1, \ldots, b\}$. An alternative is to

proceed in epochs of $b$ consecutive iterations, where at the start of each epoch the set $\{1, \dots, b\}$ is reshuffled, and $i_k$ is then selected consecutively from this ordered set. We numerically compare the convergence of both BC-RED variants in Section 5.

BC-RED updates its iterates one randomly picked block at a time using the output of $\mathsf{G}$. When the algorithm converges, it converges to the vectors in the zero set of $\mathsf{G}$

$$\mathsf{G}(\boldsymbol{x}^*) = \nabla g(\boldsymbol{x}^*) + \tau(\boldsymbol{x}^* - \mathsf{D}(\boldsymbol{x}^*)) = \mathbf{0} \quad \Leftrightarrow \quad \boldsymbol{x}^* \in \mathsf{zer}(\mathsf{G}) \coloneqq \{\boldsymbol{x} \in \mathbb{R}^n : \mathsf{G}(\boldsymbol{x}) = \mathbf{0}\}.$$

Consider the following two sets

$$\mathsf{zer}(\nabla g) \coloneqq \{\boldsymbol{x} \in \mathbb{R}^n : \nabla g(\boldsymbol{x}) = \mathbf{0}\} \quad \text{and} \quad \mathsf{fix}(\mathsf{D}) \coloneqq \{\boldsymbol{x} \in \mathbb{R}^n : \boldsymbol{x} = \mathsf{D}(\boldsymbol{x})\},$$

where $\mathsf{zer}(\nabla g)$ is the set of all critical points of the data-fidelity and $\mathsf{fix}(\mathsf{D})$ is the set of all fixed points of the denoiser. Intuitively, the fixed points of $\mathsf{D}$ correspond to all the vectors that are not denoised, and therefore can be interpreted as vectors that are *noise-free* according to the denoiser.

Note that if $\boldsymbol{x}^* \in \mathsf{zer}(\nabla g) \cap \mathsf{fix}(\mathsf{D})$, then $\mathsf{G}(\boldsymbol{x}^*) = \mathbf{0}$ and $\boldsymbol{x}^*$ is one of the solutions of BC-RED. Hence, any vector that is consistent with the data for a convex $g$ and noiseless according to $\mathsf{D}$ is in the solution set. On the other hand, when $\mathsf{zer}(\nabla g) \cap \mathsf{fix}(\mathsf{D}) = \varnothing$, then $\boldsymbol{x}^* \in \mathsf{zer}(\mathsf{G})$ corresponds to a tradeoff between the two sets, explicitly controlled via $\tau > 0$. This explicit control is one of the key differences between RED and PnP.

BC-RED benefits from considerable *flexibility* compared to the full-gradient RED. Since each update is restricted to only one block of $\boldsymbol{x}$, the algorithm is suitable for parallel implementations and can deal with problems where the vector $\boldsymbol{x}$ is distributed in space and in time. However, the maximal benefit of BC-RED is achieved when $\mathsf{G}_i$ is efficient to evaluate. Fortunately, it was systematically shown in [43] that many operators—common in machine learning, image processing, and compressive sensing—admit *coordinate friendly* updates.

For a specific example, consider the least-squares data-fidelity $g$ and a block-wise denoiser $\mathsf{D}$. Define the residual vector $r(\boldsymbol{x}) \coloneqq \boldsymbol{A}\boldsymbol{x} - \boldsymbol{y}$ and consider a single iteration of BC-RED that produces $\boldsymbol{x}^+$ by updating the $i$th block of $\boldsymbol{x}$. Then, the update direction and the residual update can be computed as

$$\mathsf{G}_i(\boldsymbol{x}) = \boldsymbol{A}_i^\mathsf{T} r(\boldsymbol{x}) + \tau(\boldsymbol{x}_i - \mathsf{D}(\boldsymbol{x}_i)) \quad \text{and} \quad r(\boldsymbol{x}^+) = r(\boldsymbol{x}) - \gamma \boldsymbol{A}_i \mathsf{G}_i(\boldsymbol{x}), \tag{8}$$

where $\boldsymbol{A}_i \in \mathbb{R}^{m \times n_i}$ is a submatrix of $\boldsymbol{A}$ consisting of the columns corresponding to the $i$th block. In many problems of practical interest [43], the complexity of working with $\boldsymbol{A}_i$ is roughly $b$ times lower than with $\boldsymbol{A}$. Also, many advanced denoisers can be effectively applied on image patches rather than on the full image [44–46]. Therefore, in such settings, the speed of $b$ iterations of BC-RED is expected to be at least comparable to a single iteration of the full-gradient RED.

## 4 Convergence Analysis and Compatibility with Proximal Optimization

In this section, we present two theoretical results related to BC-RED. We first establish its convergence to an element of $\mathsf{zer}(\mathsf{G})$ and then discuss its compatibility with the theory of proximal optimization.

### 4.1 Fixed Point Convergence of BC-RED

Our analysis requires three assumptions that together serve as sufficient conditions for convergence.

**Assumption 1.** *The operator $\mathsf{G}$ is such that $\mathsf{zer}(\mathsf{G}) \neq \varnothing$. There is a finite number $R_0$ such that the distance of the initial $\boldsymbol{x}^0 \in \mathbb{R}^n$ to the farthest element of $\mathsf{zer}(\mathsf{G})$ is bounded, that is*

$$\max_{\boldsymbol{x}^* \in \mathsf{zer}(\mathsf{G})} \|\boldsymbol{x}^0 - \boldsymbol{x}^*\|_2 \leq R_0.$$

This assumption is necessary to guarantee convergence and is related to the existence of the minimizers in the literature on traditional coordinate minimization [25–28].

The next two assumptions rely on Lipschitz constants along directions specified by specific blocks. We say that $\mathsf{G}_i$ is *block Lipschitz continuous* with constant $\lambda_i > 0$ if

$$\|\mathsf{G}_i(\boldsymbol{x}) - \mathsf{G}_i(\boldsymbol{y})\|_2 \leq \lambda_i \|\boldsymbol{h}_i\|_2, \quad \boldsymbol{x} = \boldsymbol{y} + \mathsf{U}_i \boldsymbol{h}_i, \quad \boldsymbol{y} \in \mathbb{R}^n, \boldsymbol{h}_i \in \mathbb{R}^{n_i}.$$

When $\lambda_i = 1$, we say that $\mathsf{G}_i$ is *block nonexpansive*. Note that if an operator $\mathsf{G}$ is globally $\lambda$-Lipschitz continuous, then it is straightforward to see that each $\mathsf{G}_i = \mathsf{U}_i^\mathsf{T} \mathsf{G}$ is also block $\lambda$-Lipschitz continuous.

**Assumption 2.** *The function $g$ is continuously differentiable and convex. Additionally, for each $i \in \{1, \ldots, b\}$ the block gradient $\nabla_i g$ is block Lipschitz continuous with constant $L_i > 0$. We define the largest block Lipschitz constant as $L_{\mathsf{max}} := \mathsf{max}\{L_1, \ldots, L_b\}$.*

Let $L > 0$ denote the global Lipschitz constant of $\nabla g$. We always have $L_{\mathsf{max}} \leq L$ and, for some $g$, it may even happen that $L_{\mathsf{max}} = L/b$ [28]. As we shall see, the largest possible step-size $\gamma$ of BC-RED depends on $L_{\mathsf{max}}$, while that of the full-gradient RED on $L$. Hence, one natural advantage of BC-RED is that it can often take more aggressive steps compared to the full-gradient RED.

**Assumption 3.** *The denoiser $\mathsf{D}$ is such that each block denoiser $\mathsf{D}_i$ is block nonexpansive.*

Since the proximal operator is nonexpansive [2], it automatically satisfies this assumption. We revisit this scenario in a greater depth in Section 4.2. We can now establish the following result for BC-RED.

**Theorem 1.** *Run BC-RED for $t \geq 1$ iterations with random i.i.d. block selection under Assumptions 1-3 using a fixed step-size $0 < \gamma \leq 1/(L_{\mathsf{max}} + 2\tau)$. Then, we have*

$$\mathbb{E}\left[\min_{k \in \{1, \ldots, t\}} \|\mathsf{G}(\boldsymbol{x}^{k-1})\|_2^2\right] \leq \mathbb{E}\left[\frac{1}{t}\sum_{k=1}^{t} \|\mathsf{G}(\boldsymbol{x}^{k-1})\|_2^2\right] \leq \frac{b(L_{\mathsf{max}} + 2\tau)}{\gamma t}R_0^2. \tag{9}$$

A proof of the theorem is provided in the extend version of this paper [30]. Theorem 1 establishes the fixed-point convergence of BC-RED in expectation to $\mathsf{zer}(\mathsf{G})$ with $O(1/t)$ rate. The proof relies on the monotone operator theory [47, 48], widely used in the context of convex optimization [2], including in the unified analysis of various traditional coordinate descent algorithms [49, 50]. Note that the theorem does *not* assume the existence of any regularizer $h$, which makes it applicable to denoisers beyond those characterized with explicit functions in (4) and (5).

Since $L_{\mathsf{max}} \leq L$, one important implication of Theorem 1, is that the worst-case convergence rate (in expectation) of $b$ iterations of BC-RED is better than that of a single iteration of the full-gradient RED (to see this, note that the full-gradient rate is obtained by setting $b = 1$, $L_{\mathsf{max}} = L$, and removing the expectation in (9)). This implies that in *coordinate friendly settings* (as discussed at the end of Section 3), the overall computational complexity of BC-RED can be lower than that of the full-gradient RED. This gain is primarily due to two factors: (a) possibility to pick a larger step-size $\gamma = 1/(L_{\mathsf{max}} + 2\tau)$; (b) immediate reuse of each local block-update when computing the next iterate (the full-gradient RED updates the full vector before computing the next iterate).

In the special case of $\mathsf{D}(\boldsymbol{x}) = \boldsymbol{x} - (1/\tau)\nabla h(\boldsymbol{x})$, for some convex function $h$, BC-RED reduces to the traditional coordinate descent method applied to (1). Hence, under the assumptions of Theorem 1, one can rely on the analysis of traditional randomized coordinate descent methods in [28] to obtain

$$\mathbb{E}\left[f(\boldsymbol{x}^t)\right] - f^* \leq \frac{2b}{\gamma t}R_0^2 \tag{10}$$

where $f^*$ is the minimum value in (1). However, as discussed in Section 4.2, when the denoiser is a proximal operator of some convex $h$, BC-RED is *not* directly solving (1), but rather its approximation.

Finally, note that the analysis in Theorem 1 only provides *sufficient conditions* for the convergence of BC-RED. As corroborated by our numerical studies in Section 5, the actual convergence of BC-RED is more general and often holds beyond nonexpansive denoisers. One plausible explanation for this is that such denoisers are *locally nonexpansive* over the set of input vectors used in testing. On the other hand, the recent techniques for spectral-normalization of CNNs [51–53] provide a convenient tool for building *globally nonexpansive* neural denoisers that result in provable convergence of BC-RED.

### 4.2 Convergence for Proximal Operators

One of the limitations of the current RED theory is in its limited backward compatibility with the theory of proximal optimization. For example, as discussed in [20] (see section *"Can we mimic any prior?"*), the popular total variation (TV) denoiser [31] cannot be justified with the original RED regularization function (4). In this section, we show that BC-RED (and hence also the full-gradient RED) can be used to solve (1) for any convex, closed, and proper function $h$. We do this by establishing a formal link between RED and the concept of Moreau smoothing, widely used in

nonsmooth optimization [54–56]. In particular, we consider the following proximal-operator denoiser

$$\mathsf{D}(\boldsymbol{z}) = \mathsf{prox}_{(1/\tau)h}(\boldsymbol{z}) = \underset{\boldsymbol{x} \in \mathbb{R}^n}{\arg\min} \left\{ \frac{1}{2} \|\boldsymbol{x} - \boldsymbol{z}\|_2^2 + (1/\tau)h(\boldsymbol{x}) \right\}, \quad \tau > 0, \quad \boldsymbol{z} \in \mathbb{R}^n, \qquad (11)$$

where $h$ is a closed, proper, and convex function [2]. Since the proximal operator is nonexpansive, it is also block nonexpansive, which means that Assumption 3 is automatically satisfied. Our analysis, however, requires an additional assumption using the constant $R_0$ defined in Assumption 1.

**Assumption 4.** *There is a finite number $G_0$ that bounds the largest subgradient of $h$, that is*

$$\mathsf{max}\{\|\boldsymbol{g}(\boldsymbol{x})\|_2 : \boldsymbol{g}(\boldsymbol{x}) \in \partial h(\boldsymbol{x}), \boldsymbol{x} \in \mathcal{B}(\boldsymbol{x}^0, R_0)\} \leq G_0,$$

*where $\mathcal{B}(\boldsymbol{x}^0, R_0) := \{\boldsymbol{x} \in \mathbb{R}^n : \|\boldsymbol{x} - \boldsymbol{x}^0\|_2 \leq R_0\}$ denotes a ball of radius $R_0$, centered at $\boldsymbol{x}^0$.*

This assumption on boundedness of the subgradients holds for a large number of regularizers used in practice, including both TV and the $\ell_1$-norm penalties. We can now establish the following result.

**Theorem 2.** *Run BC-RED for $t \geq 1$ iterations with random i.i.d. block selection and the denoiser* (11) *under Assumptions 1-4 using a fixed step-size $0 < \gamma \leq 1/(L_{\mathsf{max}} + 2\tau)$. Then, we have*

$$\mathbb{E}\left[f(\boldsymbol{x}^t)\right] - f^* \leq \frac{2b}{\gamma t}R_0^2 + \frac{G_0^2}{2\tau}, \qquad (12)$$

*where the function $f$ is defined in* (1) *and $f^*$ is its minimum.*

The theorem is proved in the extend version of this paper [30]. It establishes that BC-RED in expectation *approximates* the solution of (1) with an error bounded by $(G_0^2/(2\tau))$. For example, by setting $\tau = \sqrt{t}$ and $\gamma = 1/(L_{\mathsf{max}} + 2\sqrt{t})$, one obtains the following bound

$$\mathbb{E}\left[f(\boldsymbol{x}^t)\right] - f^* \leq \frac{1}{\sqrt{t}}\left[2b(L_{\mathsf{max}} + 2)R_0^2 + G_0^2\right]. \qquad (13)$$

When $h(\boldsymbol{x}) = -\log(p_{\boldsymbol{x}}(\boldsymbol{x}))$, the proximal operator corresponds to the MAP denoiser, and the solution of BC-RED corresponds to an *approximate* MAP estimator. This approximation can be made as precise as desired by considering larger values for the parameter $\tau > 0$. Note that this further justifies the RED framework by establishing that it can be used to compute a minimizer of any proper, closed, and convex (but not necessarily differentiable) $h$. Therefore, our analysis strengthens RED by showing that it can accommodate a much larger class of explicit regularization functions, beyond those characterized in (4) and (5).

## 5 Numerical Validation

There is a considerable recent interest in using advanced priors in the context of image recovery from underdetermined ($m < n$) and noisy measurements. Recent work [20–24] suggests significant performance improvements due to advanced denoisers (such as BM3D [3] or DnCNN [4]) over traditional sparsity-driven priors (such as TV [31]). Our goal is to complement these studies with several simulations validating our theoretical analysis and providing additional insights into BC-RED. The code for our implementation of BC-RED is available through the following link[1].

We consider inverse problems of form $\boldsymbol{y} = \boldsymbol{A}\boldsymbol{x} + \boldsymbol{e}$, where $\boldsymbol{e} \in \mathbb{R}^m$ is an AWGN vector and $\boldsymbol{A} \in \mathbb{R}^{m \times n}$ is a matrix corresponding to either a sparse-view Radon transform, i.i.d. zero-mean Gaussian random matrix of variance $1/m$, or radially subsampled two-dimensional Fourier transform. Such matrices are commonly used in the context of computerized tomography (CT) [57], compressive sensing [33, 34], and magnetic resonance imaging (MRI) [58], respectively. In all simulations, we set the measurement ratio to be approximately $m/n = 0.5$ with AWGN corresponding to input signal-to-noise ratio (SNR) of 30 dB and 40 dB. The images used correspond to 10 images randomly selected from the NYU fastMRI dataset [59], resized to be $160 \times 160$ pixels. BC-RED is set to work with 16 blocks, each of size $40 \times 40$ pixels. The reconstruction quality is quantified using SNR averaged over all ten test images.

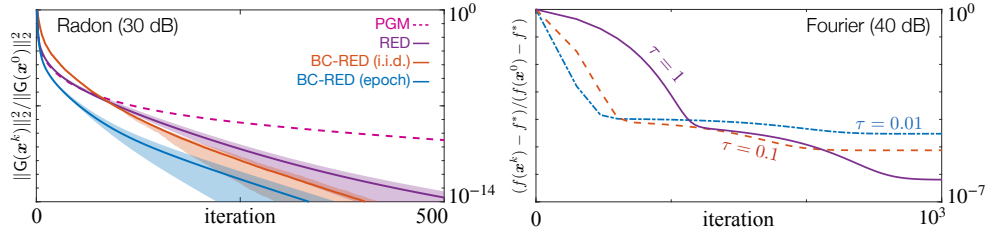

Figure 2: **Left**: *Illustration of the convergence of BC-RED under a nonexpansive DnCNN\* prior. Average normalized distance to* zer(G) *is plotted against the iteration number with the shaded areas representing the range of values attained over all test images.* **Right**: *Illustration of the influence of the parameter $\tau > 0$ for solving TV regularized least-squares problem using BC-RED. As $\tau$ increases, BC-RED provides an increasingly accurate approximation to the TV optimization problem.*

Table 1: Average SNRs obtained for different measurement matrices and image priors.

| Methods | Radon | | Random | | Fourier | |
|---|---|---|---|---|---|---|
| | 30 dB | 40 dB | 30 dB | 40 dB | 30 dB | 40 dB |
| **PGM (TV)** | 20.66 | 24.40 | 26.07 | *28.42* | 28.74 | 29.99 |
| **U-Net** | *21.90* | 21.72 | 16.37 | 16.40 | 22.11 | 22.11 |
| **RED (TV)** | 20.79 | 24.46 | 25.64 | 28.30 | 28.67 | 29.97 |
| **BC-RED (TV)** | 20.78 | 24.42 | 25.70 | 28.39 | 28.71 | 29.99 |
| **RED (BM3D)** | 21.55 | *25.24* | 26.46 | 27.82 | 28.89 | 29.79 |
| **BC-RED (BM3D)** | 21.56 | 25.16 | 26.50 | 27.88 | 28.85 | 29.80 |
| **RED (DnCNN\*)** | 20.89 | 24.38 | 26.53 | 28.05 | 29.33 | *30.32* |
| **BC-RED (DnCNN\*)** | 20.88 | 24.42 | *26.60* | 28.12 | *29.40* | *30.39* |

In addition to well-studied denoisers, such as TV and BM3D, we design our own CNN denoiser denoted DnCNN\*, which is a simplified version of the popular DnCNN denoiser (see [30] for details). This simplification reduces the computational complexity of denoising, which is important when running many iterations of BC-RED. Additionally, it makes it easier to control the global Lipschitz constant of the CNN via spectral-normalization [52]. We train DnCNN\* for the removal of AWGN at four noise levels corresponding to $\sigma \in \{5, 10, 15, 20\}$. For each experiment, we select the denoiser achieving the highest SNR value. Note that the $\sigma$ parameter of BM3D is also fine-tuned for each experiment from the same set $\{5, 10, 15, 20\}$.

Theorem 1 establishes the convergence of BC-RED in expectation to an element of zer(G). This is illustrated in Fig. 2 (left) for the Radon matrix with 30 dB noise and a nonexpansive DnCNN\* denoiser (see also [30] for additional convergence plots). The average value of $\|G(x^k)\|_2^2/\|G(x^0)\|_2^2$ is plotted against the iteration number for the full-gradient RED and BC-RED, with $b$ updates of BC-RED (each modifying a single block) represented as one iteration. We numerically tested two block selection rules for BC-RED (*i.i.d.* and *epoch*) and observed that processing in randomized epochs leads to a faster convergence. For reference, the figure also plots the normalized squared norm of the gradient mapping vectors produced by the traditional PGM with TV [60]. The shaded areas indicate the range of values taken over 10 runs corresponding to each test image. The results highlight the potential of BC-RED to enjoy a better convergence rate compared to the full-gradient RED, with BC-RED (epoch) achieving the accuracy of $10^{-10}$ in 104 iterations, while the full-gradient RED achieves the same accuracy in 190 iterations.

Theorem 2 establishes that for proximal-operator denoisers, BC-RED computes an approximate solution to (1) with an accuracy controlled by the parameter $\tau$. This is illustrated in Fig. 2 (right) for the Fourier matrix with 40 dB noise and the TV regularized least-squares problem. The average value of $(f(x^k) - f^*)/(f(x^0) - f^*)$ is plotted against the iteration number for BC-RED with $\tau \in \{0.01, 0.1, 1\}$. The optimal value $f^*$ is obtained by running the traditional PGM until convergence. As before, the figure groups $b$ updates of BC-RED as a single iteration. The results are consistent with our theoretical analysis and show that as $\tau$ increases BC-RED provides an increasingly

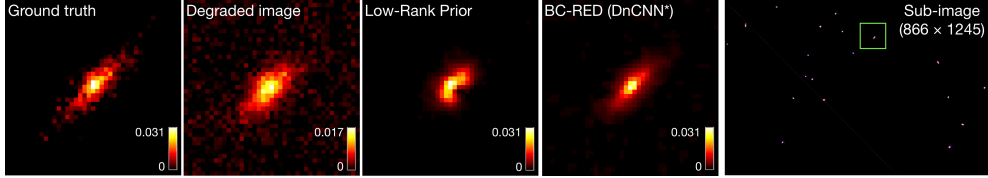

Figure 3: *Recovery of a* $8292 \times 8364$ *pixel galaxy image degraded by a spatially variant blur and a high-amount of AWGN. The efficacy of BC-RED is due to the natural sparsity in this large-scale problem, with all of the information contained in a small part of the full image.*

accurate solution to TV. On the other hand, since the range of possible values for the step-size $\gamma$ depends on $\tau$, the speed of convergence to $f^*$ is also influenced by $\tau$.

The benefits of the full-gradient RED algorithms have been well discussed in prior work [20–24]. Table 1 summarizes the average SNR performance of BC-RED in comparison to the full-gradient RED for all three matrix types and several priors. Unlike the full-gradient RED, BC-RED is implemented using block-wise denoisers that work on image patches rather than the full images. We empirically found that 40 pixel padding on the denoiser input is sufficient for BC-RED to match the performance of the full-gradient RED. The table also includes the results for the traditional PGM with TV [60] and the widely-used end-to-end U-Net approach [61, 62]. The latter first backprojects the measurements into the image domain and then denoises the result using U-Net [63]. The model was specifically trained end-to-end for the Radon matrix with 30 dB noise and applied as such to other measurement settings. All the algorithms were run until convergence with hyperparameters optimized for SNR. The DnCNN$^*$ denoiser in the table corresponds to the residual network with the Lipschitz constant of two (see [30] for details). The overall best SNR in the table is highlighted in bold-italic, while the best RED prior is highlighted in light-green. First, note the excellent agreement between BC-RED and the full-gradient RED. This close agreement between two methods is encouraging as BC-RED relies on block-wise denoising and our analysis does not establish uniqueness of the solution, yet, in practice, both methods seem to yield solutions of nearly identical quality. Second, note that BC-RED and RED provide excellent approximations to PGM-TV solutions. Third, note how (unlike U-Net) BC-RED and RED with DnCNN$^*$ generalize to different measurement models. Finally. no prior seems to be universally good on all measurement settings, which indicates to the potential benefit of tailoring specific priors to specific measurement models.

Coordinate descent methods are known to be highly beneficial in problems where both $m$ and $n$ are very large, but each measurement depends only on a small subset of the unknowns [64]. Fig. 3 demonstrates BC-RED in such large-scale setting by adopting the experimental setup from a recent work [65] (see also [30] for additional simulations). Specifically, we consider the recovery of a $8292 \times 8364$ pixel galaxy image degraded by 597 known point spread functions (PSFs) corresponding to different spatial locations. The natural sparsity of the problem makes it ideal for BC-RED, which is implemented to update $41 \times 41$ pixel blocks in a randomized fashion by only picking areas containing galaxies. The computational complexity of BC-RED is further reduced by considering a simpler variant of DnCNN$^*$ that has only four convolutional layers (see [30] for additional details). For comparison, we additionally show the result obtained by using the low-rank recovery method from [65] with all the parameters kept at the values set by the authors. Note that our intent here is not to justify DnCNN$^*$ as a prior for image deblurring, but to demonstrate that BC-RED can indeed be applied to a realistic, nontrivial image recovery task on a large image.

# 6 Conclusion and Future Work

Coordinate descent methods have become increasingly important in optimization for solving large-scale problems arising in data analysis. We have introduced BC-RED as a coordinate descent extension to the current family of RED algorithms and theoretically analyzed its convergence. Preliminary experiments suggest that BC-RED can be an effective tool in large-scale estimation problems arising in image recovery. More experiments are certainly needed to better asses the promise of this approach in various estimation tasks. For future work, we would like to explore accelerated and asynchronous variants of BC-RED to further enhance its performance in parallel settings.

## Acknowledgments

This material is based upon work supported in part by NSF award CCF-1813910 and by NVIDIA Corporation with the donation of the Titan Xp GPU for research. The authors thank B. Wohlberg as well as anonymous reviewers for insightful comments.

## Footnotes

[1]`https://github.com/wustl-cig/bcred`

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
