[Reviews · NeurIPS 2019]

Reviewer 1



A recent trend in large scale optimization, specially in the machine learning community, was to replace full gradient based algorithm by its coordinate descent counterpart. The idea being to reduce the computational cost of each iteration while enjoying similar rate of convergence. Often, the solution of maximum a posteriori (estimated with a proximal algorithm) is hard to estimate exactly when the prior is not directly available. In that case, the "proximal iteration" is replaced by "Denoised iteration" where the proximal operator of the prior is replaced by another adequate denoising operator. Such algorithm is then based on full vector update just as vanilla (proximal) gradient descent. Then, this paper follows the recent trend and make the same coordinate descent moves. Using (same) classical assumptions for establishing rate of convergence of smooth + non-smooth optimization problem, this time replacing non expansiveness of the prox by non expansiveness of the denoiser, and using same proof techniques, the authors obtain similar rates of convergence O(1/t). At the difference that, the fixed point is not explicitly known as a minimizer of some objective function. This seems to be not too surprising and immediately follows from the current results in smooth + nonsmooth convex optimization. Again, just replace the proximal operator by any other non-expansive operator and all the results remains the same (up to explicit characterization of the fixed point) When D is a proximal operator of h, the iterations in Eq. 3 correspond to the two steps 1- a (proximal) smoothing of h 2- a gradient descent on the smooth function {g + smoothed(h)}. With this view, it is natural that line 161: "when the denoiser is a proximal operator of some convex function h, BC-RED is not directly solving (1), but rather its approximation". Indeed, it solve the smoothed version of (1). These strategies are similar to Nesterov smoothing see (Beck and Teboulle, 2012: Smoothing and First Order Methods: A Unified Framework). More precisely, it seems to correspond to the "partial smoothing" method. Small questions: - Usually, proximal iteration on a smooth + nonsmooth strongly convex, enjoys geometric rate of convergence. using a general denoiser, what would be a natural assumption to get the geometric rates? - Usually, convergence analysis of bloc coordinate descent require separability of the regularizer. (If necessary) Can the authors explicitly clarify the equivalent property for a generic denoiser? Small typos: - In proof of Theorem 1, in the second and third equation after line 322, there is some missing ^2. - In the proof of Proposition 8, the function phi can be more than 1-strongly convex eg when h is already alpha-strongly convex.

Reviewer 2



EDIT: My comments about notation and figure were indeed cosmetic, hence their position at the end of the reviews. Other points have been clarified in the rebuttal, I expect the authors to include these clarifications in their manuscript despite the lack of space mentioned. ###################### The proximal gradient algorithm allows to efficiently minimize the sum of a smooth term and a non smooth regularizer. Plug and play and regularization by denoising (RED) approaches mimicks the updates of proximal gradient, using a "denoising function" instead of an explicit prox (for some cases of denoising functions, PnP and RED correspond to proximal gradient). The authors adapt this method to the case where only a subset of the parameter vector is updated at each iteration (generalizing proximal block coordinate descent in some cases) The goal is to benefit from the known speed-ups of BCD compared to full gradient. The assumptions made for convergence analysis in Thm1 sound reasonable (smoothness of the fidelity term, non expansiveness of the denoiser), but Assumption 4 is quite strong in my opinion. Thm 1 is somehow disappointing as it does not chow convergence, but only that some points get close to the zero set of G. I don't think the sentence L220 is correct, as Thm 1 does not show convergence of the iterates. I don't understand how (11) is a *generic* denoiser (L175), since it has the form of a proximal operator. ALso, in Thm 2, why not take tau going to infinity ? It seems that some constant blows up with tau, but this is not specified. The article reads well, although the reference list is quite long and often not cited very specificly, eg [5-14]. Specificities of each paper could be highlighted better in the literature review. Similar approaches have been developped in the imaging community, where the prox step (or its equivalent in ADMM) is replaced by the application of a predefined function (see eg Image deblurring by augmented Lagrangian with BM3D frame prior, Danyelan et al). The connection with these methods from another field is somehow lacking in the literature review. It is not clear for readers not familiar with PnP and RED if the algorithm corresponds to the minimization of a functional, which should be better explained. The seminal work of Paul Tseng on coordinate descent is somehow missing in the 24-27 part of the bibliography. The use of a special font for U and G is confusing in the case of U, as it looks like an union. Can't the authors use normal U ? Fig 1 is not very informative in my opinion: denoising problems are standard and the role of the function D is not particularly detailed. The sentence L17 is not completely correct to me: the true prior can be known (eg the L0 penalization for sparse vectors), but using it is intractable.

Reviewer 3



The paper is very well written, however, from my point of view, the generalisation of the study made in [21] to block coordinate approach is quite straightforward. However, Theorem 2 is a very nice and important result. But it is only a small part of the paper. It is the first time that such a bound is given and it is very useful for the understanding of RED approaches. Experimental section lake of some computation time comparison (not only with respect to the number of iterations), while block coordinate approach can be faster than a batch approach for some problem when the operator A has very correlated columns. ** After feedback ** The authors have clearly answer my concerns. For thm 1, I was convinced that a similar result was in [21], but I was wrong. So I have decided to revise my evaluation.

[Author Response · NeurIPS 2019]

We thank the reviewers for carefully reading the manuscript and providing us with valuable feedback. While the essence
of our results seem to be well understood by the reviewers, we address below some specific points that they have raised.

Note that from our perspective *the BC-RED algorithm*, *Theorem 1*, *Theorem 2*, and *the numerical experiments* are equally
important as contributions, providing new insights into using *denoisers* (including those based on *deep neural nets*) as
priors within block-coordinate estimation. Our manuscript shows that all these ingredients combine synergistically in a
novel methodology that is both theoretically rigorous and practically relevant. We made effort to give credit to all the
prior work on the topic and will include citations to all publications mentioned by the reviewers.

**Reviewer 1.** As you correctly inferred, *geometric convergence* can be obtained by strengthening *Assumption 2* to say
that $g$ is *strongly convex*. This was omitted from the submitted manuscript due to space. In the context of BC-RED,
*separable regularizers* correspond to *separable denoisers*, such as *pixel-wise* or *patch-wise* denoisers. Figure 1 (Left)
shows that while DnCNN* is *not* fully separable, it only requires 5 px padding for optimal performance. We will fix
both typos you mention in the revision. We would like to highlight two original contributions in the manuscript, namely
the infusion of *deep neural nets* into block-coordinate algorithms in a mathematically rigorous way and establishing
an explicit connection between the RED framework and nonsmooth optimization. The revised manuscript will better
explain that the traditional analysis from nonsmooth optimization does *not* simply carry over to Theorem 1, since we
assume *no objective function* (to accommodate *deep neural net* denoisers, not associated with any regularizer $h$).

**Reviewer 2.** Note that Assumption 4 holds for a large number of popular regularizers, including $\ell_2$, $\ell_1$, and TV penalties.
Theorem 1 implies that $\mathbb{E}[\|\mathsf{G}(\boldsymbol{x}^k)\|^2]$ is summable and $\mathbb{E}[\|\mathsf{G}(\boldsymbol{x}^k)\|] \to 0$, which is the best we can establish for a
*convex* $g$ and a generic denoiser. A stronger result – convergence of the iterates to a unique point $\boldsymbol{x}^* \in \mathsf{zer}(\mathsf{G})$ – can be
established when $g$ is *strongly convex*. We will clarify L220 to make this more precise. We will clarify L175 to say
that we consider a *generic* proximal operator. *No constants blow up*: it is possible to progressively take $\tau \to \infty$, as in
eq. (13), but this leads to a progressive reduction in the step-size $\gamma$ (see L187). Instead, we empirically found the benefit
of tuning $\tau$ as a free parameter. We would have loved to be more specific in citations and have a more detailed literature
review, but we were dealing with a significant space shortage (we are fully using all the 8-pages allowed by NeurIPS).
However, we will certainly include citations to both Danielyan and Tseng in the manuscript. We will clarify that in
general PnP and RED are *not* minimizing any functional. We hope that our (possibly suboptimal) notation for U and
our schematic illustrations won't preclude the reviewer from considering other merits of our manuscript. We will clarify
L17 to say that the true prior might be unknown for *certain* signals, such as natural images. We will release our code
with its documentation to GitHub after the reviews; Dropbox was used as a mechanism for anonymous code sharing.

**Reviewer 3.** While [21] is a great work, it neither analyzes block-coordinate algorithms nor provides an explicit
convergence rate. The latter is important for precisely quantifying the computational complexity of BC-RED. The
conceptual leap from the traditional RED to our analysis of BC-RED is comparable to the leap from the traditional
*gradient descent* to the Nesterov's analysis of *coordinate descent methods* [23], which is certainly not minor. We share
your enthusiasm for Theorem 2, but it will be challenging to find more space without significant revisions. We provide
some *time comparisons* in Figure 1 (Center and Right), showing that on our machine (see Section F in the supplement)
an efficient implementation of BC-RED *can* be much faster than RED, where the speed depends on the structure of the
measurement matrix and the denoiser. However, the *speed* is only one of many potential advantages of BC-RED, as it
can offer *scalability* through other mechanisms, such as effective memory management and distributed implementation.

| Padding | SNR |
|---------|-----------|
| 0 px | 27.64 dB |
| 5 px | 28.11 dB |
| 10 px | 28.12 dB |
| 20 px | 28.12 dB |
| 40 px | 28.12 dB |

Figure 1: *Left:* The performance of BC-RED for the Random matrix with 40 dB noise and *patch-wise* DnCNN*, where
the denoiser input includes an additional padding around the patch, while the output has the size of the patch. The
lower SNR for 0 px suggests *non-separability* of DnCNN*; yet, a small 5 px padding is sufficient for matching the
performance of the *full-image* DnCNN*. *Center and Right:* The convergence speed of BC-RED under *patch-wise*
DnCNN* with 40 px padding for the same setting as *Left*. Distance to $\mathsf{zer}(\mathsf{G})$ – corresponding to the *full-image* denoiser
– and SNR are plotted against time. As a reference, we provide the convergence of RED using the full-image DnCNN*
and BM3D denoisers. Since the patch-wise denoiser only *approximates* the full-image denoiser, the final accuracy of
BC-RED to $\mathsf{zer}(\mathsf{G})$ is $1.92 \times 10^{-7}$. Yet, BC-RED still matches the SNR performance of the full-gradient RED and
does this substantially faster due to its better convergence rate and reduced denoising complexity (due to patch-wise
denoising). Note also the slow convergence of RED using the full-image BM3D, due to high complexity of denoising.

[Meta-Review · NeurIPS 2019]

The authors propose a clearly written paper on extension of the Plug and Play (PnP) priors and regularization by denoising approaches. Convergence guarantees were also derived.